# Self-Powered Dual-Band Electrochromic Supercapacitor Devices for Smart Window Based on Ternary Dielectric Triboelectric Nanogenerator

**DOI:** 10.3390/nano14020229

**Published:** 2024-01-20

**Authors:** Tianxiang Zheng, Haonan Zhang, Chen Chen, Xinbo Tu, Lin Fang, Mingjie Zhang, Wen He, Peihong Wang

**Affiliations:** 1Energy Materials and Devices Key Lab of Anhui Province for Photoelectric Conversion, School of Materials Science and Engineering, Anhui University, Hefei 230601, China; b21301117@stu.ahu.edu.cn (T.Z.); b21101007@stu.ahu.edu.cn (H.Z.); b21301128@stu.ahu.edu.cn (C.C.); b21201056@stu.ahu.edu.cn (X.T.); ms22101018@stu.ahu.edu.cn (L.F.); ms2304163@stu.ahu.edu.cn (M.Z.); 2Key Laboratory of Structure and Functional Regulation of Hybrid Materials, Anhui University, Ministry of Education, Hefei 230601, China

**Keywords:** dual band, electrochromic, smart window, triboelectric nanogenerator, self-powered

## Abstract

A dual-band electrochromic supercapacitor device (DESCD) can be driven by an external power supply to modulate solar radiation, which is a promising energy-saving strategy and has broad application prospects in smart windows. However, traditional power supplies, such as batteries, supercapacitors, etc., usually face limited lifetimes and potential environmental issues. Hence, we propose a self-powered DESCD based on TiO_2_/WO_3_ dual-band electrochromic material and a ternary dielectric rotating triboelectric nanogenerator (TDR-TENG). The TDR-TENG can convert mechanical energy from the environment into electrical energy to obtain a high output of 840 V, 23.9 µA, and 327 nC. The as-prepared TDR-TENG can drive the TiO_2_/WO_3_ film to store energy with a high dual-band modulation amplitude of 41.6% in the visible (VIS) region and 84% in the near-infrared (NIR) region, decreasing the indoor–outdoor light–heat interaction and thereby reducing the building energy consumption. The self-powered DESCD demonstrated in this study has multiple functions of energy harvesting, energy storage, and energy saving, providing a promising strategy for the development of self-powered smart windows.

## 1. Introduction

In recent years, due to population growth and social development, increasing energy consumption has led to a contradiction between growing energy demand and insufficient energy supply in many countries and regions [1]. Great efforts have been devoted to alleviating the contradiction caused by the energy crisis in terms of different strategies, such as more electrical energy device savings [2,3,4,5]. The dual-band electrochromic materials can reduce the energy consumption of buildings by regulating the transmission rate of solar radiation in the visible (VIS) and near-infrared (NIR) regions and then regulating the light and heat exchange with the outside environment [1,6,7,8,9,10,11]. Meanwhile, the corresponding dual-band electrochromic supercapacitor device (DESCD) can not only regulate solar radiation but also store energy, which can improve the energy efficiency of buildings and have broad application prospects [12,13,14].

Although the DESCD has excellent performance in terms of energy savings, it inherently requires an input of external voltage [15]. Based on the concept of sustainable development, the DESCD should be driven by a sustainable green energy source. Among various sustainable energy sources (mechanical energy, solar energy, heat energy, etc.), mechanical energy comes from a wide range of sources and is almost unaffected by weather and the working environment, making it a high-quality energy source [16,17,18,19]. The triboelectric nanogenerator (TENG), which was invented by Wang’s group in 2012, has been shown to easily convert mechanical energy from the environment into electricity by coupling contact electrification and electrostatic induction [20,21,22]. Meanwhile, owing to the merits of wide material choice, simple structure, and low cost, TENG has become an ideal candidate for a sustainable green power supply [17,23,24].

Up to now, several groups have already reported their work on combining electrochromic supercapacitor devices with TENG [25,26,27]. For instance, Sun et al. [28]. integrated a piezoelectric nanogenerator, TENG, and a Ag NWs/NiO-based electrochromic supercapacitor device (ESCD) into a smart self-charging power package. The hybrid generator can produce a high output voltage of 150 V and a high output current of 20 µA. Under the impact of a human palm, the hybrid generator supplies power to the ESCD and then indicates the state of charge by changing the color of the ESCD. After that, an all-in-one self-powered power pack that combines ESCD and direct-current TENG was reported by Pu et al. [29]. Various polyaniline (PANI) patterns were used as the electrode material for ESCD. Without the assistance of complex and rigid power management circuits, the DC-TENG can collect the sliding mechanical energy on the ESCD surface. The energy storage level of the device can be displayed in real time through the color change of the polyaniline. Later on, Li et al. [30]. cleverly integrated ESCD, TENG, and organic photovoltaic into an all-weather self-powered power pack through a shared electrode layer. Organic photovoltaics can collect solar energy during the day, while the TENG can collect mechanical energy from human movement at night. The collected energy can be stored in the WO3-based ESCD. Finally, the all-weather energy storage level of the device can be visually detected through the color change of WO_3_. At present, these reported works mainly focus on the modulation of solar radiation in the visible region, while the modulation of the near-infrared (NIR) region, which has excellent performance in terms of energy savings, is rarely mentioned.

Here, we present a self-powered DESCD (SP-DESCD) through the combination of a ternary dielectric rotating TENG (TDR-TENG) and a TiO_2_/WO_3_ (TWO) composite-based electrochromic film. Without any additional power source, the self-powered DESCD could modulate the transmittance of solar radiation in the VIS and NIR regions with the change in energy storage, thus modulating the light–heat interaction between the indoor and outdoor areas. As a result, the constructed self-powered DESCD exhibits awesome apparent coloration and decolorization, a high dual-band modulation amplitude (41.6% VIS and 84% NIR), a fast response time of 21 s/3 s in the VIS region and 15 s/4 s in the NIR region, and a high output (840 V, 23.9 µA, and 327 nC). Furthermore, the indoor temperature of the house using the DESCD-based smart windows is 3.3 °C lower than that using a common FTO glass-based window. The SP-DESCD will have a wide range of applications (such as smart windows and car windows) for energy savings and green energy.

## 2. Materials and Methods

### 2.1. Materials

Tetrabutyl titanate (TBT, AR), hydrochloric acid (HCL), sodium tungstate dihydrate (Na_2_WO_4_·2H_2_O, AR), ammonium sulfate ((NH_4_)_2_SO_4_, AR), lithium perchlorate (LiCLO_4_, AR), anhydrous propylene carbonate (PC, AR), fluorine-doped tin oxide glass (FTO, 15 Ω·sq^–1^, Nippon Sheet Glass Co., Ltd., Tokyo, Japan), and all chemicals were used directly without further treatment. FTO glass was used as the base for the preparation of TWO films, which were cleaned sequentially with acetone, ethanol, and deionized water via ultrasonic cleaning for 15 min and dried in an oven before use.

### 2.2. Preparation of TWO Film

First, the TiO_2_ NW array was grown on the FTO substrate using the hydrothermal method: 0.5 mL of TBT, 15 mL of deionized water, and 15 mL of hydrochloric acid (36%) were mixed and stirred for 30 min. Then, the mixture was transferred to a Teflon-lined stainless-steel autoclave (50 mL in volume). Subsequently, the autoclave was heated in an oven at 150 °C for 2 h. Finally, the TiO_2_ NW array was obtained after washing with deionized water.

Second, WO_3_ NWs were grown on a TiO_2_ NW array through another hydrothermal method: 0.1 M Na_2_WO_4_·2H_2_O and 0.75 M (NH_4_)_2_SO_4_ were dissolved in 30 mL of deionized water and stirred. Then, a 3 M HCl solution was added dropwise to adjust the pH to around 2. Subsequently, the above-obtained TiO_2_ NW array was transferred into a sealed Teflon-lined autoclave with the conducting surface facing down. It was then heated in an oven at 200 °C for 2 h. Finally, the WTO-3 films were obtained after washing with deionized water to remove the residual reagents. The as-obtained sample was named TWO-3. For comparison, TWO-1, TWO-2, TWO-4, and TWO-5 were also prepared using the same recipe, except the molar mass of Na_2_WO_4_·2H_2_O was changed to 0.08 M, 0.09 M, 0.11 M, and 0.125 M, respectively.

### 2.3. Assembly of TDR-TENG

The TDR-TENG consists of a stator and a rotor.

The stator was assembled as below: An acrylic sheet with a thickness of 1 mm is cut into a disk with an outer diameter of 160 mm and an inner diameter of 12 mm. Meanwhile, 12 fan-shaped contour lines with an interval of 4 mm are cut with an outer diameter of 150 mm and an inner diameter of 26 mm. A 25 μm thick copper foil was pasted on its surface and then cut along the fan-shaped contour lines with a knife to form fan-shaped electrodes on the stator. The copper electrodes were divided into two groups, each of which was connected to a wire for the following test. In addition, grooves with a size of 62 × 4 mm^2^ were cut through every third sector to provide enough space for the installation of polyester fibers. The polyester fur with a length of 1 cm was fixed by passing through the grooves from the back side.

The rotor was assembled as below: an acrylic sheet with a thickness of 1 mm is cut into a disk with an outer diameter of 160 mm and an inner diameter of 8 mm. Meanwhile, 12 fan-shaped contour lines with an interval of 4 mm are cut with an outer diameter of 150 mm and an inner diameter of 26 mm. The PTFE film with a thickness of 50 μm and the Nylon film with a thickness of 25 µm were alternately pasted along the fan-shaped contour lines on the above acrylic sheet to form a fan-shaped triboelectric layer on the rotor.

The shaft with a diameter of 8 mm is locked with the rotor, the bearing is fixed with the stator, and the required TDR-TENG device is finally assembled.

### 2.4. Assembly of SP-DESCD

The pure FTO glass and the FTO glass attaching TWO films are used as the anode and cathode electrodes for DESCD, respectively. The two electrodes were adhered together using a spacer of 3M tape. Then, 1 M LiClO_4_/PC electrolyte was injected into the gap using a syringe. Finally, the DESCD was sealed with hot-melt adhesive. To obtain the self-powered DESCD, the TDR-TENG, power management circuit module, and DESCD were connected using wires. In the subsequent test, the house assembled with a self-powered DESCD-based smart window is named House I. For comparison, the common window, which is fabricated using two pieces of FTO glass and a spacer, was also fabricated. The house assembled by the common window was named House II.

### 2.5. Characterization and Measurement

The composition, crystal structure, and morphology of the materials were observed using an X-ray diffractometer (XRD, SmartLab 9 kw, Rockland, MA, USA) and a field emission scanning electron microscope (FESEM, REGULUS8230, Hitachi, Kyoto, Japan), respectively. The electrochemical tests were carried out using an electrochemical workstation (CHI760e, CH Instruments, Bee Cave, TX, USA). The optical data were recorded with an UV–Vis spectrophotometer (UV-3600, Shimadzu, Kyoto, Japan). In the motor-driven test, a stepper motor (SD60AEA06030-SC3-AP-CB, DVS Mechatronics, Tianjin, China) was employed to spin the TENG. The electrical characteristics of the TDR-TENGs were measured using an electrometer (6514, Keithley, Cleveland, OH, USA).

## 3. Results and Discussion

### 3.1. Fabrication of the TWO Composition Materials

In this study, TWO composite films were utilized as electrochromic active materials for DESCD. The preparation process of the TWO composite film is depicted in Figure 1a and the experimental section. Typically, one layer of TiO_2_ nanowire (NW) array was first grown on the surface of the FTO glass substrate using hydrothermal method. Then, with various concentrations (0.08 M, 0.09 M, 0.1 M, and 0.11 M) of precursor solution, WO_3_ NWs with different densities were grown on the TiO_2_ NW array through another hydrothermal process. As shown in Figure 1b, the strong diffraction peaks at 36.1° and 62.8° in the X-ray diffraction (XRD) pattern, which is marked as a star, correspond to the rutile TiO_2_ (JCPDS No. 21-1276), while the diffraction peaks at 13.9°, 23.3°, 28.1°, 36.8°, and 56.0° correspond to the (100), (002), (200), (202), and (204) peaks of hexagonal WO_3_ (JCPDS No. 85-2459). Meanwhile, the intensity of the characteristic peaks of WO_3_ increases with the increase in the concentration of WO_3_ precursor, indicating that the crystallinity of WO_3_ increases. The absence of the characteristic peaks of rutile TiO_2_ in the TWO may be because the WO_3_ NWs were fully covered on the TiO_2_ NW array.

To further demonstrate the morphology of the TWO composite films, SEM was performed on each sample. The TiO_2_ NWs with a diameter of 70 nm and length of 400 nm are vertically grown on the FTO glass (Figure 1c), forming a TiO_2_ NW array, which provides the breeding ground for the growth of WO_3_. As shown in Figure 1d–g, the WO_3_ NWs with interconnected network-like structures were successfully grown on the top surface of the TiO_2_ NW array. Meanwhile, the density of WO_3_ NWs increases with the increase in concentration. The influence of WO_3_ on the optical properties of composite films will be further discussed in the latter section.

### 3.2. Electrochromic and Electrochemical Properties of TWO Film

To better demonstrate the optical properties of the composite films, the transmittance spectra of pure TiO_2_ film and TWO composite films in the electrolyte of 1 M LiClO_4_/PC were tested under different voltages in the wavelength range from 350 nm to 1600 nm, where the orange dashed line of 780 nm is the boundary between the VIS region and the NIR region. The change of the NIR transmittance in the 780~1600 nm band mainly regulates the thermal radiation, while the change of the VIS transmittance in the 380~780 nm band can change the color of the material. As shown in Figure 2a, the TiO_2_ films have an average optical transmittance of more than 90% in the bleached state but have a small modulation amplitude under the stimulation of a voltage of −3 V. The TWO composite films showed more obvious optical transmittance changes at different voltage stimuli than that of pure TiO_2_ (Figure 2b–e and Appendix A), indicating that the grown WO_3_ NWs improved the optical performance of TWO composite films. However, the optical transmittance of TWO composite films in the bleached state shows a small decrease with the increase in concentration, while the optical transmittance of the colored state shows a significant decreasing trend. The modulation amplitude between the two states first increases and then decreases, presenting a normal distribution pattern. The changing trend is in line with that of the changes in the morphology of TWO composite films, which is shown in the SEM images of the previous section. This network-like structure of WO_3_ provides abundant pores, which facilitate the penetration of electrolytes and improve electrochemical kinetics. However, the excessive density leads to an increase in the ratio of optical energy reflected and absorbed during propagation, resulting in a possible decrease in the amplitude of optical modulation of the film. Among them, TWO-3 has the highest optical modulation. The optical transmittance of VIS (50%) and NIR (85%) is achieved in the bleached state, while the optical transmittance of VIS (10%) and NIR (5%) is displayed in the colored state. Both the transmittance modulation (Δ*T*) at 633 nm (VIS) and 1300 nm (NIR) show the same change trend, increasing first and then decreasing, as shown in Figure 2f. TWO-3 exhibited the highest Δ*T* of 41.6% VIS and 84% NIR. Based on the aforementioned discussion, TWO-3 was chosen as an ideal dual-band electrochromic material and was studied further in the subsequent tests.

The coloration/bleaching switching time (t_c_/t_b_) was evaluated by the time required to reach 90% Δ*T* during the coloration/bleaching process in a reversible cycle. As shown in Figure 2g, the t_c_/t_b_ of TWO-3 film can reach a fast response time of 21 s/3 s in the VIS region and 15 s/4 s in the NIR region. This is because the large number of pores and large surface area provided by WO_3_ NWs are conducive to electrolyte penetration and promote more efficient transport of Li^+^, thus speeding up the interaction between active electrode material and Li^+^. Cyclic voltammetry (CV) can be used to test the stability and reversibility of ESCD. As shown in Figure 2h and Appendix A, TWO-3 has the largest CV area. Meanwhile, the CV curves of the TWO-3 film under different sweep speeds only show changes in amplitude but not in shape, indicating that the film has good electrochemical stability. At the same time, the CV curves show several distinct oxidation and reduction peaks, indicating that the electrochromic process of TWO-3 film is accompanied by redox reactions. The galvanostatic charge–discharge (GCD) tests of TWO-3 under different current densities prove that it has the characteristics of energy storage (Figure 2i). Meanwhile, the GCD curves of pure TiO_2_ and different films are also present in Appendix A, among which TWO-3 has the largest GCD area, indicating that TWO-3 has the highest capacitance, which is conducive to the inclusion of Li^+^ ions.

### 3.3. Structure and Working Principle of TDR-TENG

TDR-TENG consists of a ternary medium of polytetrafluoroethylene (PTFE), Nylon, and polyester fur. The high output voltage can be generated through the cyclic contact separation between the three materials. The working principle of the ternary dielectric is briefly explained here according to the literature [31,32,33]. According to the triboelectric series table, the magnitude of electronegativity is PTFE > polyester fur > Nylon. Due to the difference in electronegativity, when the three dielectrics contact each other, the electrons jump from the surface of Nylon to the intermediate layer of polyester fur and then transfer to PTFE. A microscopic-level explanation is given in Appendix A, where the electron cloud barriers of the three media are modeled.

The TDR-TENG consists of a rotor and a stator, as shown in Figure 3a. PTFE film and Nylon film are alternately arranged on the acrylic substrate as a dielectric layer to form the rotor, while interdigital copper induction electrodes are pasted to the acrylic to form the stator. Polyester fur is distributed as a third medium between interdigital copper electrodes. The detailed fabrication process of TDR-TENG can be found in the experimental section. The working principle of TDR-TENG is shown in Figure 3b, where the process of charge transfer in one cycle is demonstrated. When the rotor rotates from state i into state ii, the polyester fur contacts the PTFE. Since the electronegativity of PTFE is higher than that of polyester fur, electrons from the low electronegativity of polyester fur transfer to PTFE film with high electronegativity. It results in the redistribution of surface charges to form a potential difference, which in turn generates an induced current in the external circuit that moves to the left (state ii). As the polyester skin is further transferred in state iii, more electrons from the polyester fur are injected into the PTFE film, which in turn generates an induced current in the external circuit (state iii). As the rotor continues to rotate in State iv, Nylon and polyester fur have contact with each other so that electrons from the Nylon film transfer to polyester fur. Meanwhile, the periodic charge transfer will produce an opposite current (state iv) in the external load. When the rotor continues to move to the right, it will return to state i. The polyester fur continues to accumulate charges, which will be injected into the PTFE film in the next cycle. It will cause a bigger potential difference between the PTFE film and the Nylon film. After a certain period of cycle testing, the surface charges of PTFE and Nylon gradually reach saturation.

To improve the output of TDR-TENG, the intermediate medium material has been systematically investigated. The open-circuit voltage (V_oc_), short-circuit current (I_sc_), and transferred charge (Q_sc_) of different materials are demonstrated in Figure 3c–e. Under the same conditions, the addition of the third dielectric can significantly improve output. The addition of polyester fur makes the performance of TDR-TENG better than rabbit fur. Because the electronegativity of polyester fur is inferior to that of rabbit hair, more electron transfer is generated when polyester fur contacts PTFE film.

### 3.4. Performance Optimization of TDR-TENG

To optimize the output of TDR-TENG, the number of units, spacing within tribo-layers, and rotation speed are systematically studied. As shown in Figure 4a, when the number of units is increased from two to six units, the voltage is increased from 627 V to 812 V, and the current is increased from 7.05 µA to 23.9 µA, while the charge is kept constant. The above experimental data show that the output current is proportional to the number of units when the contact area is fixed. When the number of units increases, the frequency of contact separation between frictionally charged materials increases accordingly, which leads to an increase in the charge transfer rate and, hence, an increase in current. The effect of spacing can be understood from Figure 4b, where the V_oc_, I_sc,_ and Q_sc_ of the TDR-TENG show a significant decrease as the spacing increases. Since TDR-TENG works on the principle of electrostatic induction, which is very sensitive to distance, the output decreases as the distance increases. As shown in Figure 4c, when the motor speed increases from 100 to 500 rpm, the V_oc_ increases slightly (795 V to 875 V), the short circuit current increases positively (7.88 µA to 39.5 µA), and the amount of transferred charge does not change significantly. The electrode induction area did not change, indicating that the saturated charge density did not change, so the induced charge quantity did not change. While the motor speed increases, the frequency of electrode friction and induction is improved, which correspondingly improves the current.

Based on the above experimental results, the optimized TDR-TENG can produce V_oc_ of 840 V, I_sc_ of 23.9 µA, and Q_sc_ of 327 nC under the conditions of six units, spacing of 1 mm, and rotation speed of 300 rpm. It was chosen to test the peak-to-peak output power of TDR-TENG. The peak-to-peak output power is calculated as follows:(1)$$P=I^{2}R$$
where *R* is the load resistance and *I* is the instantaneous current through the resistance. As shown in Figure 4d, the peak-to-peak output power of the TDR-TENG increases and then decreases with the rise in resistance and reaches a maximum value of 25.6 mW when the matching resistance is 100 MΩ. The charging capability of the TDR-TENG is measured by charging different capacitors. As shown in Figure 4e, the charging speed of the TDR-TENG slows down with the increase in capacitance, and it can charge a 2.2 mF capacitor to 0.3 V in 40 s. Figure 4f and Appendix A show that the TDR-TENG can easily light up 600 LEDs connected in series. To improve the efficiency of the TDR-TENG, a power management circuit (PMC) is employed. The charging speed of TDR-TENG with PMC is 14.8 times higher than that of TDR-TENG without PMC (Figure 4g). As shown in Figure 4h,i and Appendix A, the TDR-TENG with PMC can charge a 2.2 mF capacitor to 1.95 V in 48 s at a low speed of 80 rpm and keep a thermohygrometer working continuously.

### 3.5. Self-Powered System Applications

As shown in Figure 5a, the DESCD contains TWO composite films, LiClO_4_/PC electrolyte, and two FTO glasses at the bottom and top. Then, the DESCD and TDR-TENG are connected via PMC and assembled into SP-DESCD. The TDR-TENG collects high entropy energy from the environment and converts it into electrical energy, which is supplied to the DESCD with the help of the PMC to improve the charging capacity. When the DESCD is driven by an external negative voltage, electrons (e^−^) and lithium ions (Li^+^) are simultaneously injected into the lattice defect position of the WO_3_ film. The reduction reaction with the corresponding ion produces the blue product, tungsten bronze M_x_WO_3_. When WO_3_ film is applied with a positive voltage, e^−^ and Li^+^ will be withdrawn from the defect of the WO_3_ thin film, resulting in an oxidation reaction. The blue product, tungsten bronze M_x_WO_3_, will disappear, causing the color of the WO_3_ film to change from blue to colorless. This electrochromic process and principle can be described using the equation as follows:(2)$${WO}_{3}+{Li}^{+}+e^{-}↔{LiWO}_{3}$$

When the sunlight is irradiated on the surface of the DESCD, the VIS and NIR regions will be greatly attenuated by the DESCD, thus reducing the light and heat exchange between indoors and outdoors to save energy. As shown in Figure 5b,c and Appendix A, when the rotational speed is 300 rpm, the TDR-TENG can provide energy and realize the reversible transformation between the colored state and the bleached state of DESCD. After 570 s, the applied voltage reaches 3.5 V, and DESCD, with a size of 2 × 2 cm^2^, gradually changes from colorless to dark blue. When the switch is reversed, the color of DESCD fades from dark blue to colorless after about 40 s. As shown in Appendix A and the inset of Figure 5b, the DESCD with the colored state can light an LED, indicating the SP-DESCD can collect energy and power other electronic devices. To better study the energy-saving effect of the dual-band regulation of SP-DESCD, a test platform was built by using a Xenon lamp to simulate the sunlight and a motor to simulate the mechanical energy of the external environment, as shown in Figure 5d and Appendix A. The temperature changes inside the house were recorded using the K-type thermocouple. The initial temperature inside the two houses was kept at the same temperature of 28 °C. As shown in Figure 5e and Appendix A, the increasing temperature rate of House I is slower than that of House II. After 1000 s of irradiation, the temperatures in houses I and II increased to 32.7 °C and 36 °C, respectively, resulting in a temperature difference of up to 3.3 °C. The self-powered DESCD in this work can collect external mechanical energy to regulate solar radiation in VIS and NIR regions and integrates three functions of energy collection, energy saving, and energy storage, which have great application prospects in building energy conservation.

## 4. Conclusions

In summary, the SP-DESCD with triple functions of energy harvesting, energy storage, and high dual-band modulation amplitude has been successfully prepared. Without any additional power supply, the SP-DESCD can not only modulate the transmittance in the dual-band range of VIS (41.6%) and NIR (84%) but can also convert the electrical energy into chemical energy, which is stored as the color changes. It was also demonstrated that the SP-DESCD can produce V_oc_ of 840 V, I_sc_ of 23.9 µA, and Q_sc_ of 327 nC. With such output performance, it can easily light up 600 LEDs and drive a thermohygrometer under continuous operation. Under the irradiation of a Xenon lamp for 1000 s, it was demonstrated that the self-powered DESCD can reduce the light–heat exchange between the indoors and outdoors of a building. The temperature inside the house with a DESCD-based smart window achieves a difference of up to 3.3 °C compared to that with a common FTO glass-based window. This work provides a facile and effective energy-saving smart window solution that can be widely used in various buildings and automobiles.

## Figures and Tables

**Figure 1 nanomaterials-14-00229-f001:**
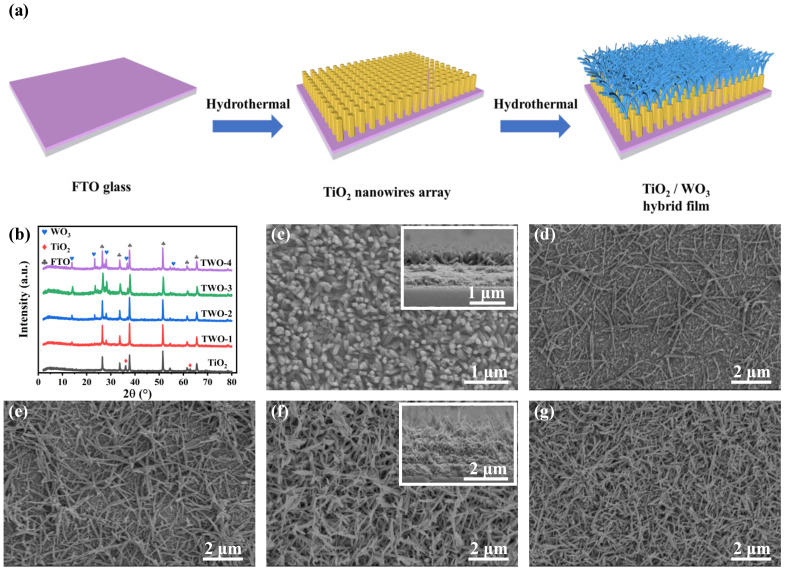
Fabrication and characterization of TWO thin films. (**a**) Schematic of the fabrication process of TWO thin films. (**b**) XRD images of TiO_2_, and different TWO films. SEM images of TWO (**c**) TiO_2_, (**d**) TWO-1, (**e**) TWO-2, (**f**) TWO-3, and (**g**) TWO-4. The insets show the corresponding cross-section image of the as-prepared sample.

**Figure 2 nanomaterials-14-00229-f002:**
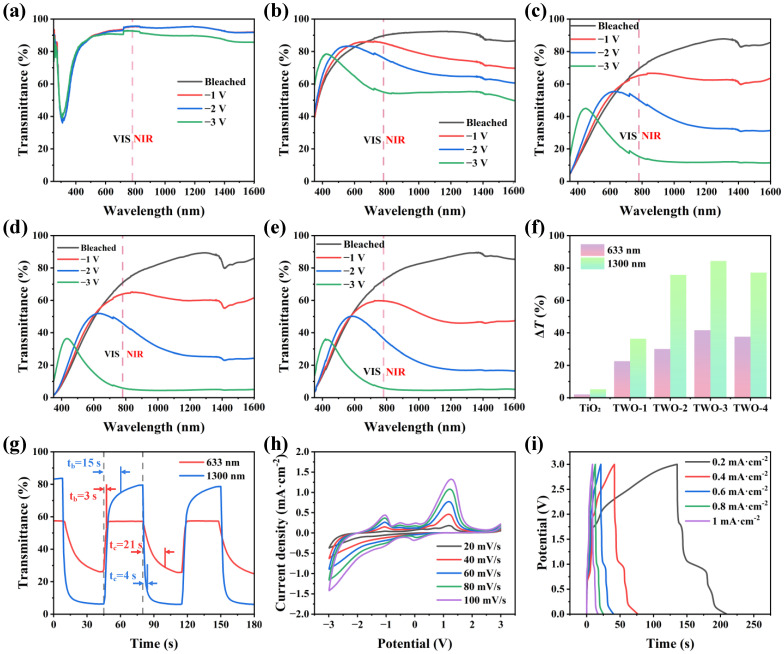
Electrochromic and electrochemical properties of TWO films. The transmittance spectra of (**a**) TiO_2_, (**b**) TWO-1, (**c**) TWO-2, (**d**) TWO-3, and (**e**) TWO-4 at different voltages in the wavelength range from 350 nm to 1600 nm. (**f**) Optical modulation amplitude of TiO_2_ and different TWO films at wavelengths of 633 nm and 1300 nm with an applied voltage of ±3 V. (**g**) Kinetic spectra of TWO-3 film under in situ transmittance at 633 nm and 1300 nm with alternating voltages of ±3 V at intervals of 30 s. (**h**) CV curves of TWO-3 film under different scan rates from 20 to 100 mV s^−1^. (**i**) GCD curves of TWO-3 films at different current densities from 0.2 to 1 mA cm^−1^.

**Figure 3 nanomaterials-14-00229-f003:**
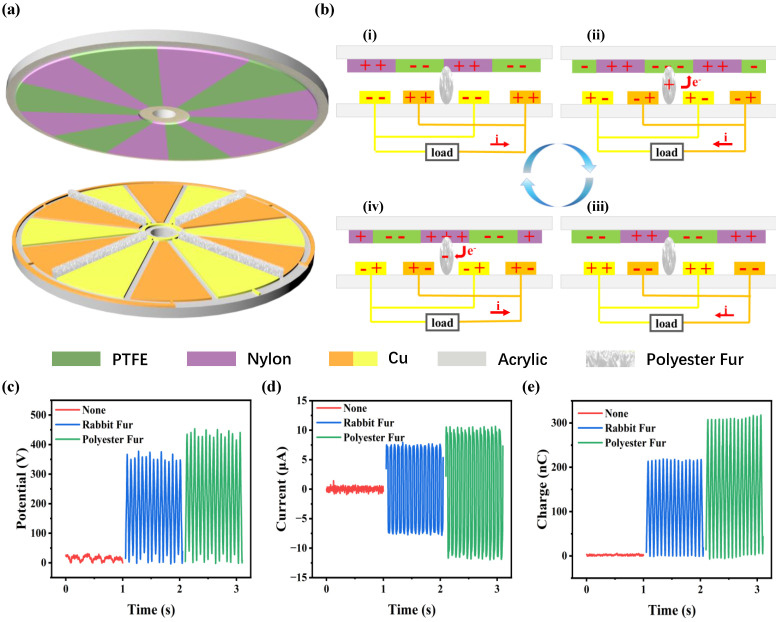
Structure and working principle of TDR-TENG. (**a**) Schematic diagram of TDR-TENG. (**b**) Schematic diagram of the working principle of TDR-TENG. The output performance of TDR-TENG with different dielectric materials. (**c**) Open-circuit voltage. (**d**) Short-circuit current. (**e**) Transferred charge.

**Figure 4 nanomaterials-14-00229-f004:**
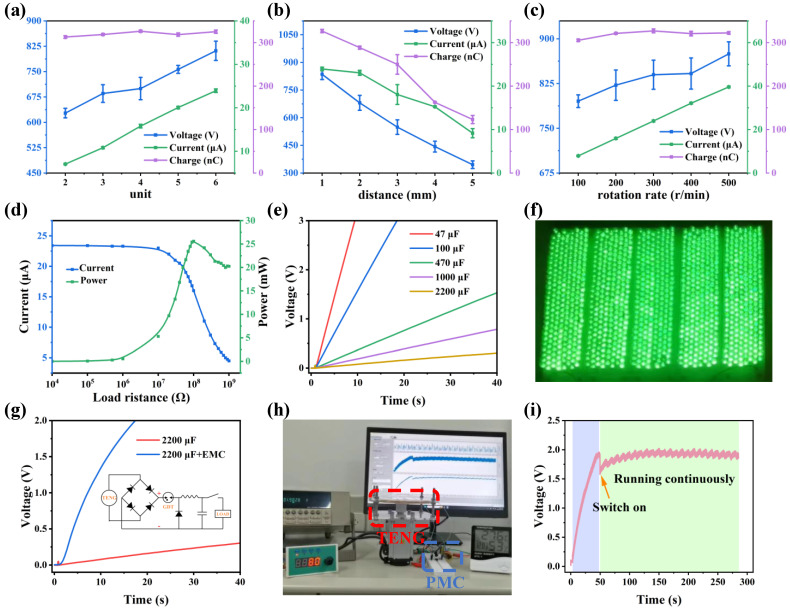
Performance and applications of TDR-TENG. (**a**) Output performance of TDR-TENG at different numbers of units. (**b**) The output performance of TDR-TENG at different spacing distances. (**c**) The output performance of TDR-TENG at different rpm. (**d**) Dependence of the peak-to-peak output power and current of TDR-TENG at a rotating speed of 300 rpm on the resistance of the external load. (**e**) Charging curves of TDR-TENG with different capacitors under 300 rpm. (**f**) A physical diagram of TDR-TENG driving 600 LEDs at a rotating speed of 300 rpm. (**g**) Comparison of the charging characteristic curve of a 2.2 mF capacitor driven by TDR-TENG after using PMC. The inset shows the circuit diagram of the PMC. (**h**) A physical diagram of TDR-TENG continuously driving a thermohygrometer at 80 rpm. (**i**) Voltage diagram of TDR-TENG continuously driving a thermohygrometer.

**Figure 5 nanomaterials-14-00229-f005:**
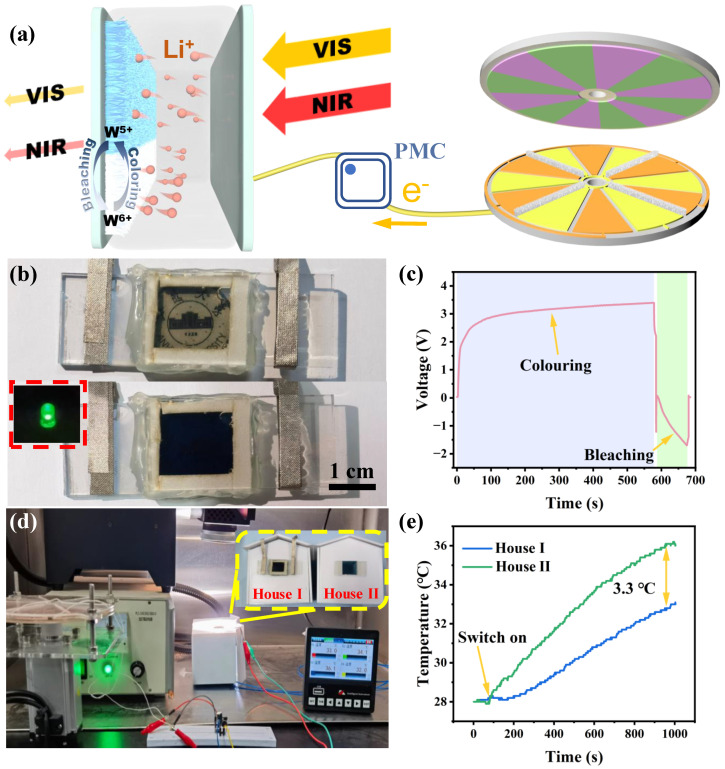
Structure and application of SP-DESCD. (**a**) Schematic structure of SP-DESCD. (**b**) Photos of self-powered DESCD with coloring state and bleaching state. The inset shows that the DESCD drives an LED. (**c**) The plots of applied voltage on the self-powered DESCD versus time. (**d**) Photos of simulated building light–heat interaction in the house with and without DESCD. (**e**) Temperature change in two houses under the irradiation of a Xenon lamp.

## Data Availability

The data presented in this study are available upon request from the corresponding author.

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
