# Peer review of "Self-Powered Dual-Band Electrochromic Supercapacitor Devices for Smart Window Based on Ternary Dielectric Triboelectric Nanogenerator"

_nanomaterials, 2024, doi:10.3390/nano14020229_

Round 1
Reviewer 1 Report
Comments and Suggestions for Authors
Dear Authors,
My comments on your manuscript are placed below.
1. The purpose of the manuscript under presentation is to address a real-world issue regarding the application of electrochemical precipitation of electrochromic layers on window glass, which offers the ability to control light transmittance and store electrical energy produced by some type of nano-generating apparatus integrated into glass.
2. The writing style of the post is understandable, and the content is engaging. However, the manuscript can be published once certain remarks have been removed. In between them are:
a. The research's main novelties are not highlighted enough, even though the Introduction should make this point quite evident.
b. TDR-TENG's design is inadequately portrayed and needs to be improved.
c. There is a lack of analytical representations in the document; the theoretical level needs to be raised. And this is considering the abundance of options this theme offers.
Technical remarks:
1. In the Introduction: “Various PANI patterns…” this abbreviation should be deciphered.
2. In the introduction: the abbreviation SP-DESCD should be deciphered
3. In Conclusions is written (840 V, 23.9 µA, and 327 nC). If it is a capacitance, it should be denoted as nF (nano-farad).
4. There are a very small number of grammar errors such as the lack of commas and the article “the” or “a” in some places. However, it can be eliminated by the technical editor.
I think that your article can be published after the elimination of remarks.
Best regards,
The reviewer
Reviewer 2 Report
Comments and Suggestions for Authors
I suggest some improvement.
Page 1: 1. Introduction: line 5: Autors should include….. “Electricity consumption and its efficiency, as well as the sustainability of cities by monitoring traffic flow, air quality and general energy consumption, has become a very important issue of modern times. Due to this increasing use of all these types of electrical devices, in the field of control and human health, any saving of electrical energy has become a very important feature of these devices, which is described in ref.:
-Improved data center energy efficiency and availability with multilayer node event processing. Energies. 2018, vol. 11, no. 9, p. 1-17. DOI: 10.3390/en11092478
Page 5: Fig. 2: This figure should bebigger.
Page 5: Last sentence: Can you comment on Optical transmittance VIS and NIR in more detail. Why is this transmittance so important and which ratio of the two would be the most optimal and what is practically achievable.
Page 9: Fig. 5b and 5d. This experimental setup (Figures) should be bigger.
Page 9: Line 12: 2.2 mF is not yet supercapacitor by meaning of value.
Reviewer 3 Report
Comments and Suggestions for Authors
Very well written and presents paper on Self-powered Dual-band Electrochromic Supercapacitor. Presentation to high standard and clear structure and novelty clearly articulated.
Fig. 1a and 3a some idea of scale
Fig 1b and 3c and 4 legend text very small
Make sure all abbreviations defined - e.g. FTO.

Comments on the Quality of English Language
English quality generally good.
Avoid non-scientific terms such as “superb”.
